# Potential of Epidermal Growth Factor-like Peptide from the Sea Cucumber *Stichopus horrens* to Increase the Growth of Human Cells: In Silico Molecular Docking Approach

**DOI:** 10.3390/md20100596

**Published:** 2022-09-23

**Authors:** Nur Shazwani Mohd Pilus, Azira Muhamad, Muhammad Ashraf Shahidan, Nurul Yuziana Mohd Yusof

**Affiliations:** 1Department of Biological Sciences and Biotechnology, Faculty of Science and Technology, Universiti Kebangsaan Malaysia (UKM), Bangi 43600, Selangor, Malaysia; 2Department of Structural Biology and Functional Omics, Malaysia Genome and Vaccine Institute (MGVI), National Institutes of Biotechnology Malaysia (NIBM), Kajang 43000, Selangor, Malaysia; 3Department of Earth Sciences and Environment, Faculty of Science and Technology, Universiti Kebangsaan Malaysia (UKM), Bangi 43600, Selangor, Malaysia

**Keywords:** sea cucumber, docking, protein–protein interaction, EGF-like, EGFR

## Abstract

The sea cucumber is prominent as a traditional remedy among Asians for wound healing due to its high capacity for regeneration after expulsion of its internal organs. A short peptide consisting of 45 amino acids from transcriptome data of *Stichopus horrens* (Sh-EGFl-1) shows a convincing capability to promote the growth of human melanoma cells. Molecular docking of Sh-EGFl-1 peptide with human epidermal growth factor receptor (hEGFR) exhibited a favorable intermolecular interaction, where most of the Sh-EGFl-1 residues interacted with calcium binding-like domains. A superimposed image of the docked structure against a human EGF–EGFR crystal model also gave an acceptable root mean square deviation (RMSD) value of less than 1.5 Å. Human cell growth was significantly improved by Sh-EGFl-1 peptide at a lower concentration in a cell proliferation assay. Gene expression profiling of the cells indicated that Sh-EGFl-1 has activates hEGFR through five epidermal growth factor signaling pathways; phosphoinositide 3-kinase (PI3K), mitogen-activated protein kinase (MAPK), phospholipase C gamma (PLC-gamma), Janus kinase-signal transducer and activator of transcription (JAK-STAT) and Ras homologous (Rho) pathways. All these pathways triggered cells’ proliferation, differentiation, survival and re-organization of the actin cytoskeleton. Overall, this marine-derived, bioactive peptide has the capability to promote proliferation and could be further explored as a cell-growth-promoting agent for biomedical and bioprocessing applications.

## 1. Introduction

Sea cucumber, locally known as gamat, has long been consumed as a food product and utilized in folk medicine among Asian and Middle Eastern communities [1]. Sea cucumber is used as traditional ointment for wound healing due to its ability to regenerate body tissues. Echinodermata organisms such as holothuria (sea cucumber), crinoidea (sea lilies), ophiuroidea (brittle starfish), asteroidea (starfish) and echinoidea (sea urchin) have the unique capability to self-regenerate body fragments as a response towards traumatic amputation. The Holothuria family, including the genus *Stichopus,* is reported to regenerate new internal organs after the evisceration or complete removal of these organs [2]. In Samoa, local people believe that consuming a sea cucumber’s internal organs provides the power of healing. After harvesting the internal organs, the Samoans throw the sea cucumber back into the sea to allow it to self-regenerate [3]. Based on scientific studies, the regeneration capacity of sea cucumber is more effective than that of sea urchins and sea stars, making it a primary regeneration model. Regeneration and restoration of its normal functions after evisceration only take a few weeks to complete [4]. This remarkable capability of tissue regeneration has been associated with extensive cell proliferation of undifferentiated cells of this organism.

There are few elements reported in sea cucumber that could contribute to its tissue regeneration property. Collagen, one of the most popular element in the cosmetic and food supplement market nowadays, is a major component of the sea cucumber’s body wall structure [5]. Docosahexaenoic acid, which is present in a large amount in the water extract of *Stichopus chloronotus,* is one of the most important fatty acids that plays a potential function in tissue repair and wound healing [6,7]. Sea cucumber is also rich in vitamins A, B1, B2 and B3, and minerals, especially calcium, magnesium, iron and zinc [8]. Even in a dried form, it is a rich source of protein containing interesting combinations of important amino acids such as glycine, glutamic acid, aspartic acid and alanine. In addition, the presence of polysaccharides, sterols, phenolics, peptides, cerebrosides, lectins and other bioactive secondary metabolites have contributed to its medicinal role in having anti-inflammatory, immunostimulatory, anti-tumor, anti-microbial, anti-hypertension, anti-angiogenic and anticancer properties [9,10,11].

Despite these nutritional properties that could be indirectly associated with the wound healing benefit of this marine organism, the full potential of sea cucumber as a source of marine therapeutic product remains unexplored and limited to the previous approaches of specific metabolite identification. Among the prospective molecules to be investigated are growth factors and cytokines, which are examples of important mitogens that ensure an effective wound healing process [12]. Several mitogens are involved directly as growth factors in cell proliferation such as epidermal growth factor (EGF), fibroblast growth factor (FGF) and insulin-like growth factor (ILGF) [13]. EGF plays important roles during wound healing, including promoting the re-differentiation of keratinocytes, and increasing the proliferation and migration of both keratinocytes and fibroblasts. It also helps in wound contraction during re-epithelization [12,14]. EGF’s roles in promoting proliferation are long proven in many in vitro studies of many types of cells such as hair follicle cells [15], amniotic epithelial cells [16] and hepatocyte-like cells [17]. Most of these cells exhibit increasing EGF activity during cell cycle at S and G2/M phases. The association between increasing DNA synthesis and the presence of EGF has been studied in a human skin model during wound healing [18]. EGF at 50 nM shows the maximum increase of DNA synthesis as compared to 0.1 nM and 10 nM EGF. The effects of EGF started to occur as early as 24 h after wounding and the mitosis effect was significantly increased on day 3 and day 7 of the treatments. In addition, Tanaka et al. [19] revealed that gelatin sheets containing EGF accelerated the wound healing process in partial thickness wounds by speeding up the re-epithelization process.

The key mechanism of cell proliferation is mitosis, which is activated through intracellular signaling pathways. Cyclin D is a critical factor during the transition of G1 to DNA synthesis phase (S phase) [20]. In eukaryotes, cells will divide when they have reached a certain size or with the presence of extracellular stimuli such as growth factors or hormones [21]. Epidermal growth factor receptor (EGFR) is a tyrosine kinase receptor that binds ligands from the EGF family, including EGF, TGF-alpha, amphiregulin, epigen, betacellulin, epiregulin and heparin-binding EGF to activate a signal transduction pathway through autophosphorylation of downstream signaling cascades. Once the EGF binds to a receptor on the cell surface, the receptor will undergo hetero or homodimerization and is internalized into the cell through endocytosis. Meanwhile, the EGF will undergo either lysosomal degradation or recycling. Several signaling pathways, such as PI3K, MAPK, MEK, STAT and Rho, are activated by EGFR, which could contribute to cell proliferation, differentiation, and migration, the reorganization of actin cytoskeleton, apoptosis and cell survival [22].

Although there were many studies that showed the effectiveness of sea cucumber extracts on wound healing and cell regeneration [23,24,25,26,27], the mechanisms involving the biomolecules that might play significant roles in these biological processes are yet to be explored. In our study, we have identified contig 498513 that was retrieved from our transcriptomic data of *S. horrens*. The contig was among a few lists identified with epidermal growth factor-like human domains. We predicted a protein structure based on the DNA contig sequence and simulated molecular docking of the structure with human EGFR [28]. Based on the docking simulation, the 4.9 kDa peptide showed inter-residue contact with human EGFR. The peptide, named Sh-EGF-like 1 or Sh-EGFl-l, was synthesized and was assessed for its interaction with human receptors based on a ligand binding assay. The effect of Sh-EGFl-l on human melanoma cell proliferation was analyzed and was validated using a PCR array for any interaction of genes involved in human EGF/PDGF pathway.

## 2. Results

### 2.1. Selection of Sh-EGFl-1 Peptide from Contig 498513 Protein Sequence

ORFinder analysis predicted that contig 498513 sequences consist of a 378 bp coding region. The length of the predicted protein is 128 amino acids and has 57% identity with EGF of the purple sea urchin, *Strongylocentrotus purpuratus*. A domain search for this protein showed the presence of a calcium-binding EGF-like superfamily in two different locations (Figure 1). Attachment of calcium ion in this domain helped stabilize its interaction with another protein through the N-terminal domain [29]. In order to determine the active residues from the Sh-EGFl-1 protein, which directly interact with human receptors, two approaches were applied as follows: (i) one based on a literature search on the 1IVO model, and (ii) one based on CPORT software prediction [30]. Based on the 1IVO crystal structure of EGF–EGFR complex reported by Ogiso et al. [31], 48% (10 of 21 residues) were listed in the calcium-binding-like domain region of *S. horrens* predicted protein i.e., Asn77, Cys79, Phe80, Ser81, Ser82, Pro83, Cys84, Cys90, Cys99 and Thr106. In addition, CPORT predicts passive residues, which contributed to the interaction, but the scores were not penalized when either of the residues were not included during docking. Overall, there were 42 passive residues predicted by CPORT including Met5, Leu7, His9, Phe10, Ile11, Ser12, Cys15, Lys16, Glu25, Lys26, Leu30, Asn75, Ile76, Glu78, Glu85, Asn86, Gln87, Gly88, Ile89, Gln91, Asp92, Glu93, Gly96, Tyr97, Asn98, Val100, Cys101, Gln102, Gly104, Phe105, Gly107, Thr108, His109, Glu111, Ser112, Leu115, Asn116, His117, Val118, Leu122, His123 and Cys124. Out of these 42 residues, 23 (55%) were in the second calcium-binding domain which were used for this modeling. When the *S. horrens* protein was compared to EGF in the 1IVO human crystal structure [31], 30 residues were aligned between Cys84 and Glu111, which happened to be located in the second calcium-binding domain (Appendix A). BLASTP version 2.7.1 analysis revealed that residues from this domain have 31% identity similarity with EGF 1IVO structure, based on E-value of 2 × 10^5^ and 73% sequence coverage [32]. Based on the protein docking with human EGFR, a similar domain of the *S. horrens* protein showed a good binding interaction (Appendix A). Therefore, a short amino acid sequence from residues 75 to 111 of the protein sequence was selected and named as Sh-EGFl-1.

### 2.2. Modeling of Sh-EGFl-1 Peptide

The Sh-EGFl-1 peptide sequence consisting of 45 amino acids was submitted to I-TASSER for 3D structure prediction [33]. Structure refinement was performed using ModRefiner and MolProbity [34,35]. Based on PROMOTIF prediction, the peptide has four strands, two beta hairpins, seven beta turns, one gamma turn and three disulphide bonds. A molecular dynamic simulation was performed to investigate the stability of Sh-EGFl-1 peptide interaction with human EGFR. The model reached an equilibrium state in the last 50 nanoseconds (ns) of the simulation, based on the average deviation of the RMSD value of the backbone atoms, which was relatively 1 Å (Appendix A).

Further comparison with EGF 1IVO showed that there were eight conserved residues in Sh-EGFl-1. Three disulphide bonds and conserved residues of cysteine and glycine were found inside the globular structure of both structures (Figure 2).

### 2.3. Molecular Docking Model between Sh-EGFl-1 Peptide and Human EGFR

Docking structures suggested by HADDOCK were scrutinized based on several parameters such as the HADDOCK score, binding affinity, dissociation constant, Z-score, van der Waals forces and electrostatic energy [36]. Table 1 shows the values of the docked complex structures produced by HADDOCK. The most reliable docked model of Sh-EGFl-1 to human EGFR was chosen to compare with 1IVO.

In terms of orientation, Sh-EGFl-1 was found to have a similar docking orientation to 1IVO EGF when docked to domains I and III of EGFR, except that the beta sheet rotated 90^o^ clockwise (Figure 3). Active residue analysis in other protein kinase families revealed that residues such as glycine, leucine and glutamic acid contributed to the hydrogen bond strength towards protein kinase primary chains, including cyclin-dependent kinase 2 (CDK2) and fibroblast growth factor receptor 2 (FGFR2). Similar to glycine, its small size causes the interaction to be more flexible when forming hydrogen bonds with the ligand [37]. According to Bissantz et al. [38], the ideal distance between a hydrogen acceptor and donor is 2.8 to 3.1 Å. In this study, the docking of Sh-EGFl-1 peptide with human EGFR was estimated at 2.95 Å distance, which is close to the average length of an 1IVO complex, 2.98 Å. Meanwhile, the RMSD value between the HADDOCK docking structure of Sh-EGFl-1 and 1IVO is less than 3.0 Å (0.852 Å), which makes it an acceptable docking model. According to the Critical Prediction of Interactions (CAPRI), the quality of a docking structure relies on ligand–RMSD or l–RMSD values; values less than 10 Å is acceptable, less than 5 Å is intermediate while less than 1 Å is the best [39]. This value is calculated based on the distance of the core hydrogen atom, alpha carbon, carbon, nitrogen and oxygen between the two ligands of the models being compared, as well as based on the overlap and the alignment of receptors. In this study, the value obtained for Sh-EGFl-1 peptide with the human receptor docking model is 7.873 Å, which falls within the acceptable range.

According to 1IVO crystal structure [31], the EGF binding site is located in domains I and III of EGFR. Asn32 interacts with Gln16 in domain I while in domain III of EGFR, Arg41 interacts with Asp355 of EGFR and Tyr13 interacts with Phe357 of EGFR. When EGFR of the same location was examined, the model given by HADDOCK resulted in no interaction at Asp355 but there was a hydrophobic interaction between Phe357 and Gly14, while Gln16 may form a hydrogen bond with Thr34 of the Sh-EGFl-1 but at more than 3 Å distance. However, there are three EGFR residues that simultaneously formed hydrogen bonds and hydrophobic interactions with Sh-EGFl-1 residues. Two of the residues (Thr15 and Tyr45) were found in domain I and another residue (Ser418) is located in domain III. Based on Figure 4, Thr15 formed a hydrogen bond with Cys25 and at the same time formed a hydrophobic interaction with Val26. Tyr45 formed a hydrogen bond with Gly33 and concurrently formed a hydrophobic interaction with Pro29. On the other hand, in domain III, Ser418 formed a hydrogen bond with Asn1 and simultaneously formed a hydrophobic interaction with Asp21.

After investigating the crystal structure of human EGF in 1IVO, the main residues contributing to the interaction of the peptide with the receptor are Asn32 interacting within domain I of EGFR and Tyr13 and Arg41 within domain III of EGFR. The benzene ring of the aromatic amino acid, Tyr13, seems to stack upon the phenyl ring of another aromatic amino acid, Phe357, in domain III of EGFR. At the same time, Arg41 interacts through a salt bridge with Asp355 in EGFR while simultaneously forming van der Waals forces with Tyr13 and Phe357. Meanwhile, Asn32 formed a hydrogen bond with Gln16 in the domain I EGFR. The importance of these residues in receptor binding was revealed in a mutation study where the replacement of Tyr13 with Val, Ile, Ala and Arg reduced the binding activity up to more than 90% while replacement with Phe or Leu retained 75% of the binding affinity [40]. However, in the HADDOCK model, Tyr13 in 1IVO was replaced by Pro9, Asn32 was replaced by Val26, and Arg41 was replaced by His35 (Figure 5). All three residues formed hydrophobic interactions with residues of the human receptor. The replacement of proline with tyrosine could still allow the Sh-EGFl-1 peptide to interact with human receptor because both residues have aromatic side chains, while histidine and arginine have positive-charged side chains. Moreover, it is postulated that Thr15 in EGFR enhances the ligand binding of Sh-EGFl-1 peptide by forming simultaneous interactions with two of Sh-EGFl-1’s residues: a hydrophobic interaction with Val26 as well as a hydrogen bond with Cys25.

Kuo et al. [41] previously reported the dissociation constant (K_d_) value of EGF and the EGFR complex, obtained experimentally using a surface plasmon resonance (SPR) and atomic force microscopy (AFM), which was approximately 1.77 × 10^−7^ M. However, a lower K_d_ value was obtained for Sh-EGFl-1 and human EGFR (Table 1). The values are different because docking is based on the scoring functions used. However, docking of both human EGF and Sh-EGFl-1 peptide to EGFR showed almost similar values. A lower K_d_ value means dissociation occurs at a slow rate and the binding affinity (K_a_) between these molecules is strong. Both K_a_ and K_d_ values are important to determine how fast a ligand dissociates from its receptor is because the effect of a reaction to reach the equilibrium is different depending on types of molecules, and K_d_ determines the prolonged effect of bound molecules [42].

HADDOCK appears to be a promising approach, as docking is assisted by given data either from interface restraints of Nuclear Magnetic Resonance (NMR), mutagenesis experiments or bioinformatics. The success of the HADDOCK model prediction is reflected in recent CAPRI experiments and based on the number of structures calculated using the software deposited to PDB [43]. Hydrogen bonding and hydrophobic interactions affect the affinity binding energy between ligand and receptor. These two types of bonding combined create a stronger bond when compared to hydrogen bonding alone. Hydrophobic interactions stabilize ligands during binding interphase [44]. A hydrogen bond forms between two electronegative atoms when hydrogen atoms act as donors and covalently attach to the acceptor atom, such as nitrogen and oxygen, which carries free electron pairs. Hydrogen bonds are mostly found in ligand–protein interactions, protein folding as well as in enzyme catalytic reactions [45]. Upon protein binding, the active residues undergo quick desolvation allowing the ligand to enter an active site and replace bulk water molecules by forming hydrogen bonds with the ligand atom at a permissible energy [37].

### 2.4. Sh-EGFl-1 Peptide–EGFR Binding Assay

An experiment was performed to test the ability of Sh-EGFl-1 peptide interaction with human receptor. Ligand binding assay, adapted from King et al. [46] was performed with modifications using human cells. Both Sh-EGFl-1 and rhEGF were first conjugated with N-hydroxysuccinimidobiotin (NHS-biotin) and were added to human cells. Any binding of peptide to EGFR was detected through the reaction of NeutrAvidin conjugated with horseradish peroxidase enzyme and 2,2′-azino-bis-3-ethylbenzothiazoline-6-sulfonic acid (ABTS) as substrate.

The graph in Figure 6 shows the absorbance readings at 405 nm, which was due to the Sh-EGFl-1 and rhEGF binding reaction to the human receptor at 10, 50 and 500 nM. The trend of the absorbance reading initially was similar for both peptides; there was an increment at 10 nM of both peptides. However, for Sh-EGFl-1, a concentration of 500 nM seems to promote interactions with EGFR the most, whereas for rhEGF, 10 nM is the optimum concentration for binding. Interaction of Sh-EGFl-1 with the human receptor increased by almost 10% when 500 nM Sh-EGFl-1 was introduced, as compared to rhEGF, where the binding signal only increased 4% although the peptide concentration had been increased ten times. The calculation for the relative binding percentage is available in Appendix A Appendix A.

A similar ligand binding study was observed by King et al. [46] in which the binding of biotinylated EGF to receptors, at concentrations ranging from 0.156 to 5 ng/mL, shows consistent increases in the absorbance reading. The increasing binding activity was determined by both the concentration of biotinylated EGF tested and the number of cells used. In this study, inoculating about 80,000 cells gives a reproducible result as opposed to when lower number of cells was used. Increasing the ligand concentration could improve assessment of the binding signal of Sh-EGFl-1 peptide with human EGFR since the expected K_d_ value during the plate assay is probably higher than the K_d_ value obtained by Kuo et al. [41] through SPR and AFM.

### 2.5. Effect of Sh-EGFl-1 Peptide on Cell Proliferation and Cell Morphology

The number of the cells cultured in medium containing 10 nM Sh-EGFl-1, 50 nM Sh-EGFl-1 and 50 nM rhEGF was significantly higher than that cultured in a serum-free medium. Figure 7 shows that only cells cultured in a medium containing 10 nM Sh-EGFl-1 exhibited increased numbers until the end of the experiment on day 4. Although it shows a stable increment throughout the experiment, it is considered as not significantly different compared with the serum-free treatment. Interestingly, 50 nM for both recombinant human EGF and Sh-EGFl-1 shows a temporary increment of cell numbers that lasted for 48 h. The cell numbers in both treatments dropped drastically right after the medium change on the next day onwards. This is probably due to the over dosage of growth factor, given that it causes cells to lose their adherent property. As reported by Broecker et al. [47], increased an EGF concentration induced a loss of cell adhesion as the cell loses the integrin-mediated signaling receptor, which is important for extracellular cell binding to the flask surface. This eventually leads to cell cycle halting and contributes to apoptosis in addition to suppressed proliferation activity.

Based on morphological observations, slight changes of cell shape were observed in most of the treatments during serum starvation, when cells started to lose adherence (Figure 8). However, cells treated with 10 nm Sh-EGFl-1, 50 nM Sh-EGFl-1 and 50 nM rhEGF showed better recovery as the cells were elongated when they attached to the flask surface. Nevertheless, changing the medium after 48 h caused most of the cells to lose attachment from the flask except in the presence of 10 nM Sh-EGFl-1.

### 2.6. Effect of Sh-EGFl-1 on EGF Pathway

The EGF pathway induces the autophosphorylation of tyrosine kinase to activate many signaling induction pathways, promoting cell proliferation, metabolism, protein growth, differentiation and migration. The main purpose of EGFR activation via EGF binding is to provide enough signal for cells to migrate from G1 state to S phase of the cell cycle [22]. The key player of this signal is cyclin D that binds to CDK4/6 to phosphorylate protein retinoblastoma [48,49]. Phosphorylated Rb releases E2F transcription factor to participate in the transcription of cyclin E, thus leading to G1/S progression. There are a few pathways that induce cyclin D expression including via ERK MAPK and AKT signaling pathways [50,51].

Based on the heat map diagram (Figure 9), cells treated with 10 nM Sh-EGFl-1 showed an upregulation of most genes involved in EGF/PDGF pathways. Gene expression profiling of cells in serum-free medium as a negative control (NC) showed a similar pattern with cells in 10 nM Sh-EGFl-1. However, cells in the 10 nM rhEGF showed the opposite gene expression profile. Cells treated with Sh-EGFl-1 may be involved in various activities, including angiogenesis, due to a relatively high expression of MAPK8, MAPK9, MAP2K7, MKK, MAPK1, *ELK1* and c-JUN. A high expression of Src, Ras, Kras and Raf indicates the involvement of cells in migration, while Shc, Fos, IKK, MAP2K1 and EGF are related to proliferation. Gene activation of PLCG1, GAB1, PIK3R1, GSK3B, AKT1/2/3 and MEK is linked to differentiation, while Gr2, EGFR, PIK3R2, EGF, STAT5A and DUSP are involved in the cell survival. Dual-specificity phosphatase enzyme (DUSP) is reported to be an important protein in limiting the intensity and duration of EGF in keratinocytes, thus acting as a positive feedback to this signaling [52]. Overall, Sh-EGFl-1 has activated three signaling pathways: PI3K, MAPK and Rho.

The PI3K signaling pathway regulates metabolism, proliferation, cell size, survival and motility while the MAPK signaling pathway prevents apoptosis and initiates proliferation, differentiation and cell migration [22]. Rho signaling pathway is activated to ensure cells retain their shape through actin cytoskeleton rearrangement while migrating and undergoing mitosis [53]. Based on the known MAPK signaling in mammalian species, there are three common pathways involved depending on its stimuli. Growth factors induce RAS–RAF–MEK–ERK1/2, stresses induce RAC–MEKK–MKK4/7–JNK1/2/3 and inflammatory cytokines induce the RHO–MEKK1–MKK3/6-p38 pathway [54]. Hence, looking at the increased expression of RAS, RAF, MEK1/2 and ERK1/2, we concluded that Sh-EGFl-1 does have an impact as a growth factor that activates MAPK through the ERK1/2 pathway. Cells treated with Sh-EGFl-1 might not respond to proliferation due to inflammation because IL-2 is minimally expressed. It is acceptable that no wound was introduced during this experiment as wounds trigger lymphocyte to release lymphokines such as IL-2, which bind to heparin sulphate and help T-lymphocytes to proliferate [55].

Based on the qPCR array result (Table 2), Sh-EGFl-1 is hypothesized to interact with human EGFR. This interaction could probably initiate autophosphorylation by recruiting adaptor proteins such as GRB2, SHC, NCK and STATs to the receptor [56]. The binding of adaptor protein to EGFR could lead to various signaling pathways. As an example, GRB2 is the main component for EGFR signaling to activate RAS. The GRB2 domain can either bind directly to EGFR or through adaptor protein SHC. Based on the increased expression of SHC in the qPCR array, it is most likely that SOS binds to SHC to activate RAS. Activated RAS will induce either RAF kinase or PI3K. The observed increase in gene expression of MEK1/2 and ERK1/2 indicates that RAF probably has been induced by RAS to initiate a cascade of genes to undergo phosphorylation and activate MEK1/2 and ERK1/2. Next, ERK1/2 has possibly assisted in the transcription and translation of cyclin D in the cell nucleus through the formation of the AP-1 complex. Increased *ELK1* expression is predicted to transcribe *c-FOS* since the *c-FOS* gene is also increased in the qPCR array. Then, *c-FOS* could interact with c-JUN to form the AP-1 complex. This complex, together with c-MYC, most likely induced the transcription of cyclin D in the nucleus [57].

Another possible route of Sh-EGFl-1 in promoting cell proliferation is by inducing protein docking via GAB1 to activate PI3K. PI3K may have converted PIP2 to PIP3 to allow the phosphorylation of *AKT* before activating PDK1. Upregulation of the *AKT* gene may result in phosphorylation of NF-kappa B to ensure cell survival [58]. *AKT* could also phosphorylate other substrates such as GSK-3 until GSK-3 expression is high enough to suppress cyclin D production. However, suppression of TSC2 by AKT and mTOR activation may occur based on the high expression of p70S6K and EIF4E. The activation of mTOR is hypothesized to suppress 4E-BP so that cyclin D translation will occur with the help of translation initiation factor EIF4E [22].

Sh-EGFl-1 could activate STAT1 and STAT3 by phosphorylating both molecules to form complexes with JAK1 and JAK2 as intermediates. After forming the complexes, STAT1 and STAT3 move to the nucleus to start transcription. STAT3 may be induced without JAK as intermediate, in which STAT3 is activated after Sh-EGFl-1 binds to EGFR through c-Src. Besides that, Sh-EGFl-1 binding to EGFR could also activate the protein oncogene Vav, a guanine nucleotide exhcange factor for the GTPase Rho family to activate the Rho signaling pathway for actin cytoskeleton rearrangement. Vav is expected to induce RAC to activate the JNK pathway [59]. Lastly, Sh-EGFl-1 binding most probably induced NCK binding to the CBL domain, thus activating PAK1. This is due to increased expression of MEKK1 and MKK4/7, which may have been induced by JNK through PAK1. Activated JNK then entered the nucleus and phosphorylated the transcription factor *c-FOS* and c-JUN to initiate cyclin D transcription.

## 3. Discussion

Even though many animal and human studies have verified the benefits of sea cucumber for wound healing, the mode of action through the molecular mechanism of this therapeutic effect is not yet fully understood. Literature search revealed that most of the bioactive compounds of sea cucumber could be involved during the first wound healing process, particularly the inflammation phase [23,24,60,61]. For instance, unsaturated fatty acids such as EPA and DHA are known to aid blood clotting, arginine helps to increase the number of lymphocytes while saponin acts as surfactant during phagocytosis [7,10,62,63]. Nonetheless, the eradication of debris and microbes, as well as stabilization of reactive oxygen species during this phase, are deemed useful for progression to the subsequent healing phases, hence, accelerating the healing process [64].

The ability of the sea cucumber to undergo complete tissue regeneration within three weeks has stimulated an interest to study this unique species. Extensive cell proliferation is the backbone to the activity of cell molecules involved during tissue regeneration. Wound healing processes in humans can be related to the sea cucumber’s ability to regenerate organs. The basic processes for regeneration are cell dedifferentiation, trans-differentiation, and both proliferation and migration, which are also part of the wound healing mechanism [65]. Based on RNA sequencing analysis on the regeneration of the sea cucumber’s organ after evisceration, there are several biological processes that are being identified daily. These processes consist of wound healing (day 7 till 14), blastema formation (day 3 till 7), lumen formation (day 7 till 14), intestine regeneration (day 14 till 21) and, lastly, growth of intestine cells (day 21 until fully recovered). The genes that are identified to be highly expressed are those related to extracellular matrix, muscle tissue regeneration and the three signaling pathways for Wnt, BMP and EGF [66]. All these three pathways are important in regulating the proliferation, differentiation, survival and apoptosis of cells, similar to the process in humans [4,67].

Many studies reported that EGF binding to the EGFR binding site caused the joining of two EGFR molecules at domains II and IV, and the process is known as dimerization [31,68,69]. Before EGF binds to EGFR at domains I and III, the site is initially exposed, where it looks like a C-shape structure in which domains II and IV are holding onto each other through intramolecular interactions. Once EGF binds, the structure changes, disturbing the interaction between domains II and IV thus exposing the domains to form a heterodimer [70]. However, the changes in domains II and IV demand a higher EGF binding affinity energy to domains I and III beforehand to overcome these intramolecular interactions [71]. After EGF binds to EGFR, the internalization of receptors occurs to activate the cascades. At this moment, EGF molecules on the cells will deplete while the number and size of cell vesicles will increase and move towards the cell perinuclear area as a preparation for endocytosis [72]. The internalization process is a prominent EGFR regulation because it determines whether a cell needs to continue the signaling according to its pathway [73]. If Cbl or an adaptor protein like Shc is upregulated this indicates that there is an interaction of EGF and EGFR, thus causing the internalization of receptor into the cells [74].

The rationale of testing peptide concentrations at 10 nM was based on a study done by Clark et al. [75]. In the study, the EGFR expression in carcinoma cells was prolonged for 10 h when 10 nM of EGF was exerted, as compared to only 4 h of EGFR expression in 0.4 nM EGF. This is supported by another experiment by Zhao et al. [76] where cell proliferation of CHO cells can only be induced when exerted with EGF concentrations below 0.1 ng/mL. In contrast, cell proliferation at higher EGF concentrations (1 to 100 ng/mL) was suppressed. The process of EGF binding to its receptor and the internalization of receptor took only less than 30 min to accomplish, according to an experiment on HeLa cells [77]. However, a previous study has reported that a prolonged ERK activation is required for cell transition from G1 to S phase for proliferation purpose [78]. Based on cell proliferation assays and morphological observations along with the qPCR assay after 4 h of peptide interventions, Sh-EGFl-1 could have interacted with human receptors due to increased expression of the adaptor proteins Shc, Sos and Grb2. Sh-EGFl-1 has the ability to sustain a prolonged EGFR effect in melanoma cells because the cells are able to maintain their adherent properties, as well as survive and proliferate, albeit rather slowly. In fact, the interaction of 10 nM Sh-EGFl-1 is more preferrable than recombinant human EGF at the same concentration. However, we suggest that qPCR analysis at the time intervals of 24, 48 and 96 h along with cell proliferation experiments should be done in the future to track the gene expression changes of the cells.

EGF is among the growth factors that play an important role during this process. In this study, we screened from our previous *S. horrens*’s genomic data (unpublished yet) and narrowed down the Sh-EGFl-1 peptide as potential biomolecule to be related to wound healing or tissue regeneration. Based on solid evidence of the molecular docking analysis, the interaction of Sh-EGFl-1 with human receptors was further assessed by ligand binding assays and its role in the EGF pathway was verified through a qPCR array analysis.

A relatively high expression of STAT3, c-Jun and *c-FOS* indicated that the cells were in the process of transcription of cyclin D. The relationship of these genes with wound healing has been proven by the fact that when STAT3 and AP1 complexes formed by c-Jun and *c-FOS* genes were removed, wound healing was delayed [79]. During wound healing, EGF induced cells to proliferate so that they are actively replacing the torn epithelium cells [12]. In another experiment, EGF promotes the re-epithelization of keratinocyte cells by activating EGFR expression but EGFR expression was diminished during chronic wound healing, causing worsening pathogenesis [64]. EGF also activated fibroblast and keratinocyte migration by regulating components of the extracellular matrix such as fibrillin, collagen and matrix metalloproteinase to support tissue remodeling during the third phase of the healing process [80].

## 4. Materials and Methods

### 4.1. S. horrens Protein Modeling and Interaction Study through Molecular Docking

Contig 498513 with 1093 bp was derived from transcriptomic data of *S. horrens*. The sequence was submitted to ORFinder webserver for prediction of its coding sequence. After obtaining the predicted coding sequence, the identity of the protein was identified using Blastp. A domain homology search was also performed. The predicted protein sequence was submitted to I-TASSER webserver for the prediction of its tertiary structure. The predicted structure was refined using ModRefiner to get a structure similar to the native protein structure in terms of hydrogen bonds, protein backbone topology and the positioning of side chains. The structure was further refined using MolProbity where the number of hydrogen atoms were checked or added if needed, at the right position. The final structure was checked based on the available protein structure in PDBSum database using EMBL-EBI webserver. The structure of *S. horrens* protein was analyzed using PROMOTIF to obtain the structure summary, including the number of alpha helices, beta sheets, strands, hairpins and disulphide bridges. PROCHECK program was run to obtain a summary in the form of an image through Ramachandran plot, in which the unnecessary residues are shown to give an idea of the structure quality (results are shown in Appendix A). Human EGF and EGFR structures were obtained from crystal structure 1IVO in the PDB database. The heterodimer form of EGF–EGFR consists of four chains; chains A and D make up EGFR while chains B and C make up EGF. We edited the structure to become a homodimer where only chain A and chain C were chosen as reference in our study. Bulk water molecules were deleted to ease docking. The active residues for human EGFR (chain A) were retrieved from the literature review of the 1IVO structure. The active residues for *S. horrens* protein were searched using CPORT webserver. All the active residues identified were used as inputs during the docking simulation. The docking simulation of *S. horrens* protein and human EGFR was performed using HADDOCK version 2.2 webserver. The WeNMR grid-enabled server was utilized to perform the docking simulations. The docked complex structures produced by the HADDOCK were based on the Z-score, therefore the cluster with the lowest Z-score value was chosen as the most reliable docked structure of the peptide to the receptor. Once finished, the binding affinity and K_d_ values of the docked structure were predicted using PRODIGY webserver [81]. A docking comparison of *S. horrens* protein with 1IVO was performed by aligning both models using PyMol version 2.3.2. The LigPlot+ version 2.1 program was employed to analyze the docking results [82]. The structure orientation, interacting residues, and number and types of bonding were carefully inspected. The values of RMSD were also calculated and compared (results are shown in Appendix A).

### 4.2. Sh-EGFl-1 Peptide Modeling and Interaction Study through Molecular Docking

As a result of *S. horrens* protein’s interaction study, a 45 amino acid sequence was selected (NINEC FSSPC ENQGI CQDEI DGYNC VCQPG FTGTH CESSM LNHVI) and named Sh-EGFl-1 peptide. The modeling and the docking processes for this short peptide sequence were repeated as for the process done with *S. horrens* protein. The tertiary structure of the peptide was predicted using I-TASSER webserver. Structure refinement was made using ModRefiner then MolProbity. The refined model was accessed by PROMOTIF and PROCHECK to obtain more detailed information, including the number of alpha helices, beta sheet, loops, hairpins and disulphide bonds. A molecular dynamics simulation was carried out on the refined peptide model using YASARA version 21.8.27 for 100 ns [83]. AMBER14 force field was used in the production run and RMSD values of the backbone atoms were monitored.

All the amino acids of Sh-EGFl-1 were selected as active residues whereas the active residues for human EGFR (chain A) were retrieved from the literature review of 1IVO structure. All the active residues identified were used as inputs during docking. The docking simulation of Sh-EGFl-1 and human EGFR (chain A) was performed using HADDOCK2.2 webserver. The WeNMR grid-enabled server was utilized to perform the docking simulations and the cluster produced with the lowest Z-score value was chosen for further analysis. Once finished, the binding affinity and K_d_ values of the docked structure were predicted using PRODIGY webserver. Docking comparisons of Sh-EGF-1 peptide with 1IVO were performed by aligning both models using PyMol version 2.3.2. The LigPlot+ version 2.1 program was employed to analyze the docking results. The structure orientation, interacting residues, and number and types of bonding were carefully inspected. The values of ligand–RMSD were also calculated and compared.

### 4.3. Preparation of Peptide

A short peptide sequence derived from the protein translation of Contig 498513, was submitted to order. The 4.9 kDa Sh-EGFl-1 peptide (H-NINEC FSSPC ENQGI CQDEI DGYNC VCQPG FTGTH CESSM LNHVI-OH) was synthesized by Mimotopes Pty. Ltd., Australia through a Fmoc solid-phase peptide chemistry method. A master stock concentration of 1 mg/mL was prepared by diluting lyophilized peptide with PBS buffer pH 7.4. Recombinant human EGF (rhEGF) (H-MNSDS ECPLS HDGYC LHDGV CMYIE ALDKY ACNCV VGYIG ERCQY RDLKW WELR-OH) was purchased from Thermo Scientific. For the cell proliferation assay and morphology observation experiments, peptides were added fresh into the serum-free growth medium to concentrations of 10 nM and 50 nM. For the binding affinity assay experiment, biotinylation of Sh-EGFl-1 and rhEGF peptides with NHS–biotin (Sigma Aldrich, Burlington, MA, USA) was performed according to the manufacturer’s instructions. The diluted peptides in PBS pH 7.4 were changed into carbonate buffer 0.1 M carbonate buffer pH 9.5 (0.07 M sodium bicarbonate, 0.03 M sodium carbonate) for an optimal biotinylation environment. A 10 X volume of cold acetone was added to the peptides and vortexed for 15 s. The mixture was incubated for 30 min at −20 °C before being centrifuged at 16,100× *g* for 24 min at 4 °C. The supernatant was discarded and the pellet was air-dried and fully dissolved in carbonate buffer pH 9.5. A total of 20 mg/mL of NHS–biotin was prepared in 1 mL DMSO. A volume of 0.13 µL and 0.1 µL of NHS–biotin was added to the Sh-EGFl-1 and rhEGF peptides, respectively. Both reaction mixtures were vortexed for a while and incubated in a thermoblock with 400 rpm shaking speed for 4 h at room temperature. Any unattached biotin molecules were sieved out using a PD-10 desalting column containing Sephadex G-25 medium (GE Life Sciences, Chicago, IL, USA). PBS buffer pH 7.4 was used as eluent to collect the conjugated peptides out from the column. The concentration of the biotin-conjugated peptides was determined using a NanoDrop spectrophotometer (Thermo Scientific, Waltham, MA, USA) and BCA assay.

### 4.4. Cell Culture

Human melanoma cells (CRL-1872) were purchased from ATCC, Rocksville, MD, USA. Cells were cultured in a complete growth medium, which consisted of Minimum Essential Medium (MEM) (Gibco, Waltham, MA, USA) supplemented with 10% fetal bovine serum (FBS) (Invitrogen, Waltham, MA, USA) and 1% antibiotic–antimycotic (Invitrogen, Waltham, MA, USA), in a humidified incubator at 37 °C, equilibrated with 5% CO_2_. Cells were subcultured every two days until reaching confluency in T25 flask (TPP, Trasadingen, Switzerland) before being seeded in designated plates for specific experiment objectives.

### 4.5. Cell Proliferation Assay and Cell Morphology Observation

Cells were seeded at 80,000 cells per well in a 24-well plate containing 400 µL of complete growth medium. After 24 h of culture, the medium was removed, washed once with PBS, replaced with serum-free medium and left to culture for another 16 h. The medium was removed and replaced with corresponding treatment medium, each in triplicates; 10 nM and 50 nM of Sh-EGFl-1, 10 nM and 50 nM of rhEGF were used as a positive control and serum-free medium was used as a negative control. Cells were cultured in a single plate to track their growth every 24 h during the four days of experiment. Medium was changed every two days to ensure that cells did not lack nutrients. A crossed line was made at the bottom of each well to mark the four cell-counting areas that were recorded daily using a microscope digital camera. Cells were counted manually based on the images visualized using ImageJ version 2.0.

### 4.6. Binding Affinity Assay

Melanoma cells were seeded at 90% confluency in a 96-well plate and cultured in a humidified incubator at 37 °C, equilibrated with 5% CO_2_ until the next day. When all cells were attached to the surface, the plate was placed on ice for a while to reduce the protein metabolism rate, so the cells were not easily detached. The medium was removed and cells were washed twice using 200 µL of cold PBS buffer. A volume of 100 µL cold formalin was added to the cells to enhance cell attachment to the surface. The plate was placed on ice for the first 5 min, then continued for another 15 min incubation at room temperature. Cells were washed twice using 200 µL of cold PBS. A total of 100 µL of melanoma growth medium (MEM basal medium and 3% FBS) was added and cells were incubated at 37 °C for 30 min. The medium was discarded and replaced with 100 µL of treatment medium (MEM basal medium supplemented with 0.1% BSA, 200 mM HEPES buffer and Sh-EGFl-1 and rhEGF at concentrations of 10, 50 and 500 nM respectively). Cells were incubated at 37 °C for 1 h. The medium was discarded, and cells were washed twice using 200 µL cold PBS and 0.3 M NaCl. A total of 100 µL horseradish peroxidase (HRP) conjugated with NeutrAvidin solution (Thermo Scientific, Waltham, MA, USA) was added with a concentration of 1:10,000 onto cells and incubated at 37 °C for 1 h. The medium was discarded, and cells were washed twice with 200 µL of cold PBS and 0.3 M NaCl. A total of 50 µL ABTS (Thermo Scientific, Waltham, MA, USA) was added as a substrate for peroxidase. The plate was incubated at room temperature for 25 min and the reaction was stopped by adding 50 µL of 1% SDS. An absorbance reading at 405 nm was recorded using microplate reader (Thermo Scientific, Waltham, MA, USA). The experiment was run in triplicates.

### 4.7. Gene Expression Analysis of EGF Pathway

Cells were seeded at 80,000 cells per well of 24-well plates in 400 µL of complete growth medium. Cells were grown in MEM basal medium supplemented with 10% FBS and 1% antibiotic–antimycotic in a humidified incubator at 37 °C, equilibrated with 5% CO_2_. After 24 h of culture, the medium was removed, washed once with PBS, replaced with serum-free medium and left to culture for another 16 h. Medium was removed and replaced with corresponding treatment medium each in triplicates; 10 nM and 50 nM of Sh-EGFl-1, 10 nM and 50 nM of rhEGF were used as a positive control and serum-free medium was used as a negative control. Cells were harvested after 4 h. Medium was removed and 200 μL of Trizol reagent (Thermo Scientific, Waltham, MA, USA) was immediately added.

### 4.8. RNA Extraction

RNA extraction was performed according to Trizol reagent manufacturer’s protocol. An elution volume of 15 μL RNase-free water was added before the concentration and purity were determined using NanoDrop.

### 4.9. cDNA Synthesis

A total of 100 ng RNA was used initially for the cDNA synthesis reaction. The reaction was carried out according to RT^2^ First Strand Kit’s protocol (Qiagen, Venlo, Netherlands). Genomic DNA was removed according to the manufacturer’s protocol prior to cDNA strand generation.

### 4.10. Quantitative Real-Time PCR Containing Array of EGF Pathway Genes

The 10 μL cDNA of human skin cells was diluted with 91 μL prior to qPCR reaction. Primer pairs of 84 genes involved in the EGF pathway, five housekeeping genes such as ribosomal protein large, P0 (RPL0), glyceraldehyde-3-phosphate-dehydrogenase (GAPDH), beta-actin (ACTB), beta-2-microglobulin (B2M) and hypoxanthine phosphoribosyltransferase 1 (HPRT1), one positive control gene for genomic DNA contamination, three reverse transcription control genes and three PCR-positive controls were included in the Qiagen PAHS-040ZA qPCR array plate (Qiagen, Venlo, Netherlands). A list of all genes analyzed in this study is available in Appendix A Appendix A. The mixture reaction was prepared according to the protocol and run using BioRad CFX96TM. Samples of 10 nM rhEGF and 10 nM Sh-EGFl-1 were prepared in duplicates. The following thermal profile was applied: 1 cycle at 95 °C for 10 min, 40 cycles at 95 °C for 15 s, 60 °C for 1 min, with the ramp rate from 95 °C to 60 °C was set at 1 °C every second. The data were then analyzed through Qiagen web portal. Data for serum-free medium were selected as a control group and the differences in gene expression between control and treatment groups were calculated using the 2^(−delta delta C_T_) method. The difference, known as fold change, was obtained from the delta delta C_T_ values, which are the genes of interest and the average housekeeping genes, followed by the delta delta C_T_ (delta C_T_ (treatment group)—delta CT (control group)).

## 5. Conclusions and Future Perspective

Our study revealed that Sh-EGFl-1 activates the EGFR pathway and induces cell proliferation via PI3K–AKT–GSK3, Ras–Raf–MEK–ERK–MAPK, PLC gamma, STAT and Rho signaling pathways. Among the suggestions that can be done to further this study is to test the effect of Sh-EGFl-1 on gene expressions involved in the EGF/PDGF pathway during a longer cell culture period. Also, a comparative analysis of gene expression can be performed to see the development of EGFR phosphorylation activity in relation to time. In silico prediction of EGF-like peptide is an advantage of this research because it can be synthesized for use in any biopharmaceuticals application i.e., as a serum substitute [28,84,85], and also in medicine i.e., during chronic wound healing [12,86]. Combining Sh-EGFl-1 with other growth factors such as FGF or PDGF may speed up the wound healing process as they work synergistically to trigger more pathways [87,88,89].

Overall, this work successfully answered one of many biological questions regarding the therapeutic effect of *S. horren’s* wound healing, based on the role of Sh-EGFl-1 to increase cell proliferation. As a unique marine organism that has the ability to regenerate, there are many more useful biomolecules from Asia’s sea cucumber species that could be explored for various applications in life sciences.

## Figures and Tables

**Figure 1 marinedrugs-20-00596-f001:**
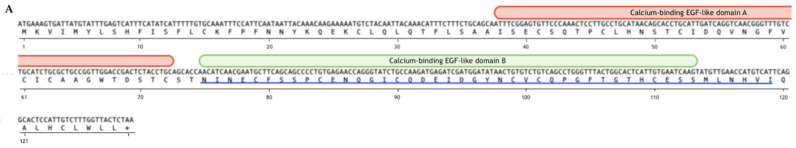
Structure of the predicted protein according to the sequence of a contig derived from transcriptome data of *S. horrens*. (**A**) The sequence of the 378 bp coding region is shown in the upper line while the predicted protein sequence of the 128 amino acids is shown in the bottom line. Calcium-binding EGF-like domains were present in two locations, domain A from 38 to 72 amino acids and domain B from 75 to 111 amino acids. The blue line at the bottom of the predicted protein sequence is the Sh-EGFl-1 peptide sequence used in this study. (**B**) Three-dimensional structure of the predicted protein with calcium-binding EGF-like domains A and B are shown in red and green, respectively. The three-dimensional image was generated by Maestro (Schrödinger Release 2022-3: Maestro, Schrödinger, LLC., New York, NY, USA, 2021).

**Figure 2 marinedrugs-20-00596-f002:**
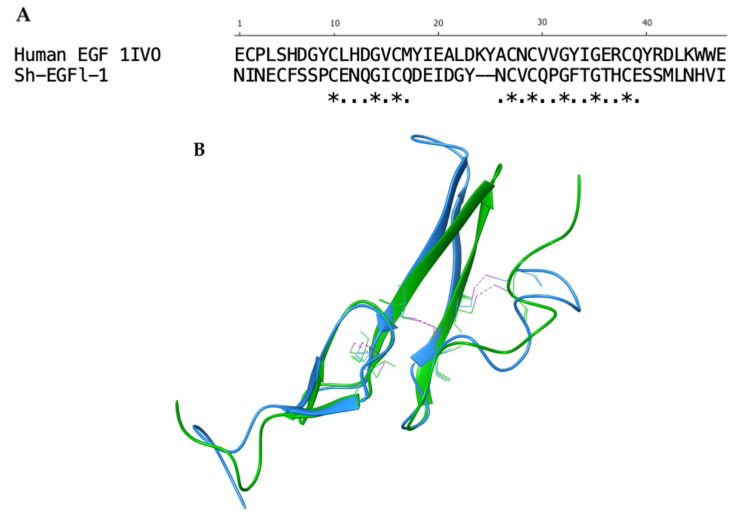
Alignment of Sh-EGFl-1 peptide and human EGF 1IVO. (**A**) Sequence alignment generated after aligning the three-dimensional structure of Sh-EGFl-1 with human EGF 1IVO. The asterisks ‘*’ show the fully conserved residues while the dots ‘.’ show the partially conserved residues. (**B**) Three disulphide bonds (pink) were found in both Sh-EGFl-1 (green) and human EGF 1IVO (blue). The three-dimensional image was generated by Maestro (Schrödinger Release 2022-3: Maestro, Schrödinger, LLC, New York, NY, USA, 2021).

**Figure 3 marinedrugs-20-00596-f003:**
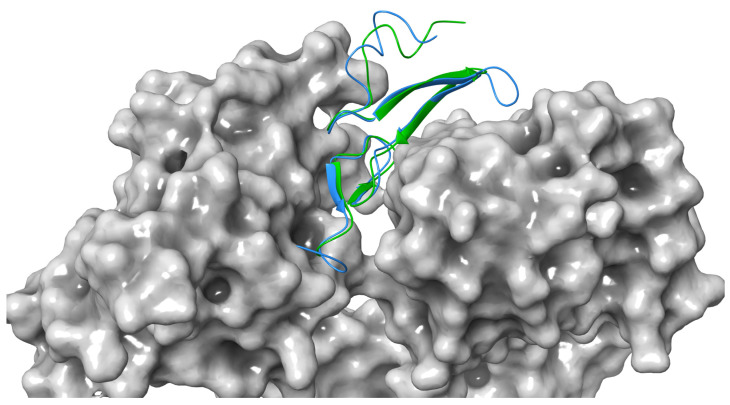
Docking comparison between Sh-EGFl-1 peptide (green) and human EGF 1IVO (blue) against human receptor (grey). The three-dimensional image was generated by Maestro (Schrödinger Release 2022-3: Maestro, Schrödinger, LLC, New York, NY, 2021).

**Figure 4 marinedrugs-20-00596-f004:**
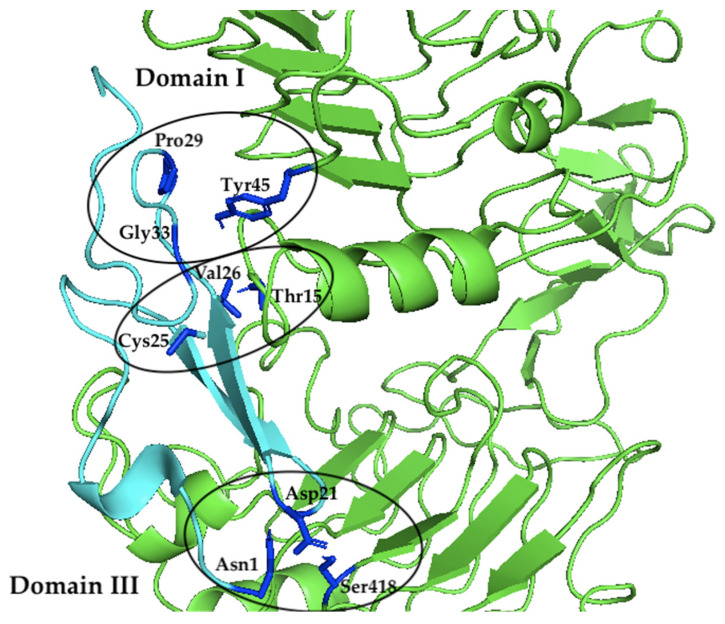
The interacting residues of Sh-EGFl-1 (cyan) within domains I and III of EGFR (green). In domain I of EGFR, Thr15 formed a hydrogen bond with Cys25 and concurrently formed a hydrophobic interaction with Val26. Tyr45 formed a hydrogen bond with Gly33 and at the same time formed a hydrophobic interaction with Pro29. In domain III of EGFR, Ser418 formed a hydrogen bond with Asn1 and simultaneously formed a hydrophobic interaction with Asp21. The image was generated using PyMol version 2.3.2, Schrödinger, LLC, New York, NY, USA).

**Figure 5 marinedrugs-20-00596-f005:**
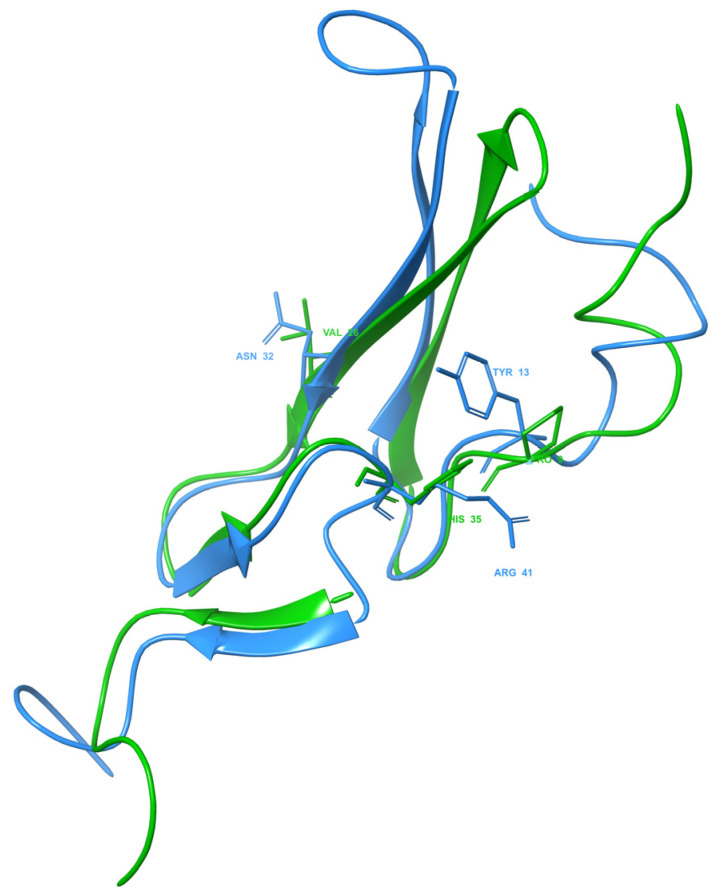
Three key residues of Sh-EGFl-1 peptide (green) and EGF 1IVO (blue). In the EGF 1IVO structure, Tyr13, Asn32 and Arg41 are the important residues in EGFR binding. After aligning the structure with the Sh-EGFl-1 peptide, the same residue positions were replaced by Pro9, Val26 and His35, respectively. The three-dimensional image was generated by Maestro (Schrödinger Release 2022-3: Maestro, Schrödinger, LLC, New York, NY, USA, 2021).

**Figure 6 marinedrugs-20-00596-f006:**
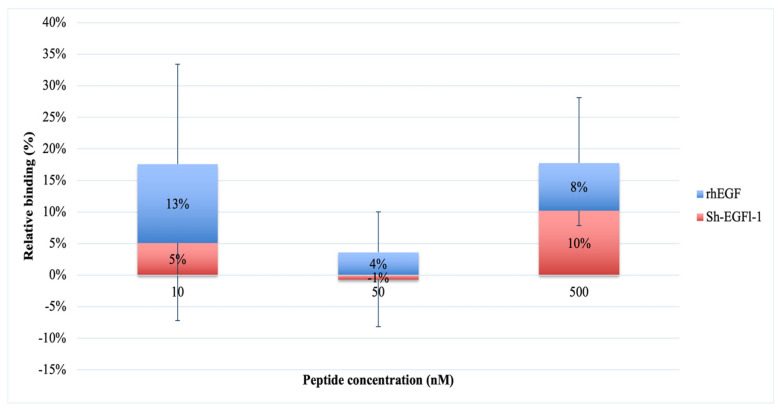
Interaction of Sh-EGFl-1 peptide with human EGFR represented by a relative binding percentage based on the absorption reading at 405 nm as a result of a chained reaction of biotinylated EGF with NeutrAvidin and ABTS.

**Figure 7 marinedrugs-20-00596-f007:**
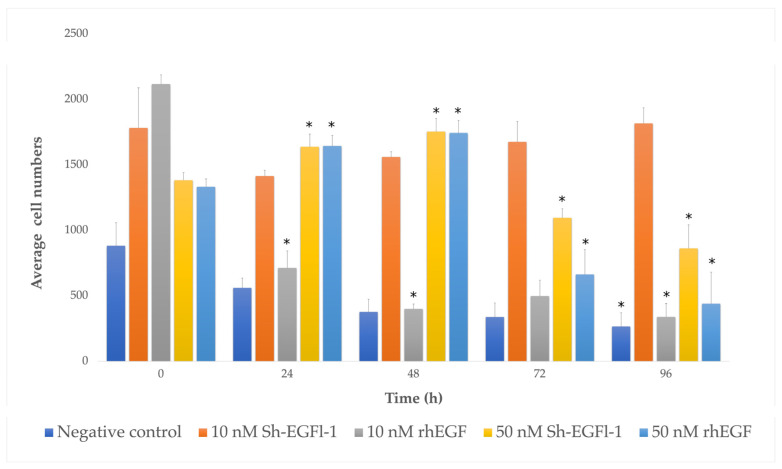
Human cells growing, as represented by bar charts, in different treatment media according to days. The symbol “*” shows that there is a significant difference of cell numbers in each treatment medium depending on the culture duration (*p* < 0.05, *p* = 0.00).

**Figure 8 marinedrugs-20-00596-f008:**
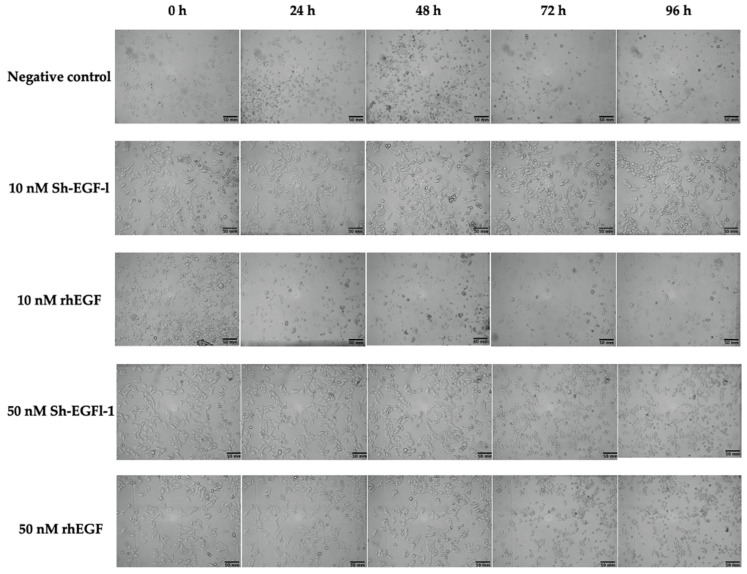
Cell morphology according to treatment medium during four days of observation. Images were taken at 10× magnification.

**Figure 9 marinedrugs-20-00596-f009:**
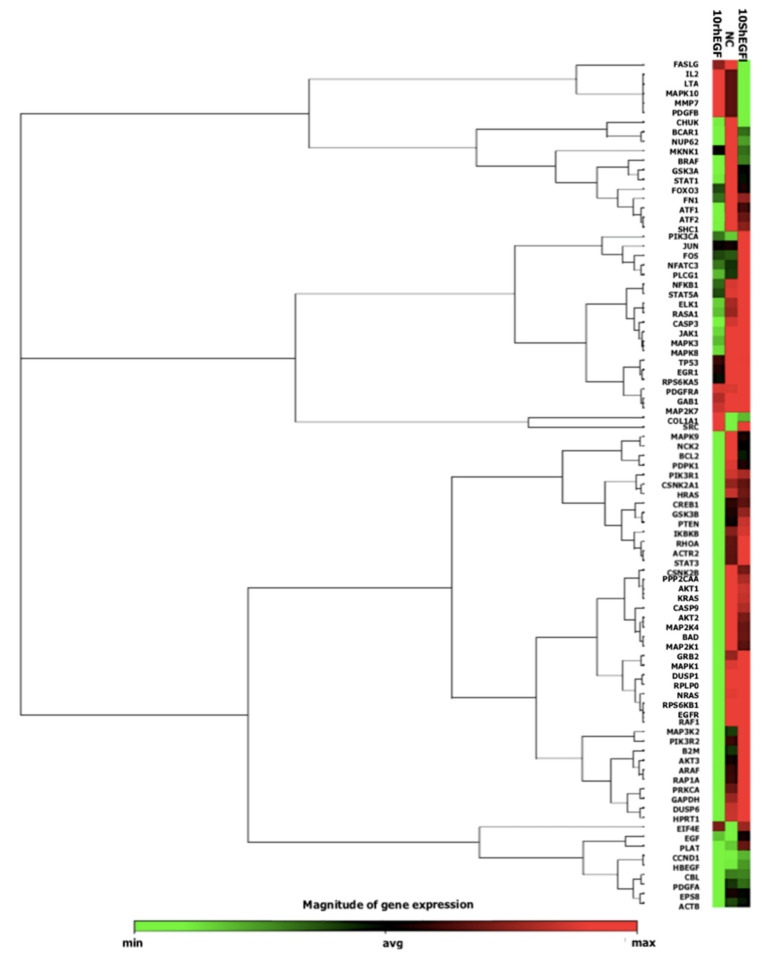
A heat map showing expression from the qPCR array containing genes related to the EGF/PDGF pathway activated by Sh-EGFl-1 and recombinant human EGF (rhEGF).

**Table 1 marinedrugs-20-00596-t001:** Interaction profile analysis of Sh-EGFl-1 with human EGFR in 1IVO based on HADDOCK simulation. Comparison was made against interaction of EGF–EGFR 1IVO.

Docking Results	Human EGF-EGFR (1IVO)	Sh-EGFl-1-Human EGFR
Binding affinity (kcal/mol)	−15.4	−15.1
Dissociation constant, K_d_ (M)	5.2 × 10^−12^	7.8 × 10^−12^
Number of interacting residues in EGF	25	23
Number of hydrogen bonds	12	10
Number of residues aligned with EGF 1IVO	-	8
Number of residues aligned with EGFR 1IVO	-	14
RMSD (Å) (<3 Å)	-	0.852
Ligand-RMSD (Å) (<10 Å)	-	7.873

**Table 2 marinedrugs-20-00596-t002:** Expression profile of treatment medium containing Sh-EGFl-1 and rhEGF against serum-free medium as the negative control, according to the signaling pathways activated by EGFR.

	10 nM Sh-EGFl-1		10 nM rhEGF		Negative Control
MAPK signaling					
CBL	−	(1.00)	−	(0.96)	−
SHC1	+	(0.94)	−	(0.80)	+
HRAS	+	(0.93)	−	(0.62)	+
RASA1 (p120GAP)	+	(1.07)	−	(0.83)	+
RAP1A	+	(1.14)	−	(0.80)	unc.
ATF1	+	(0.04)	−	(0.87)	+
ATF2	+	(0.89)	−	(0.67)	+
NFATC3	+	(1.22)	−	(0.96)	−
CASP3	+	(1.05)	−	(0.70)	+
CASP9	+	(0.93)	−	(0.63)	+
MAP2K1 (MEK1)	unc.	(0.91)	−	(0.74)	+
MAP2K4 (MKK4)	unc.	(0.84)	−	(0.51)	+
MAP2K7 (MKK7)	+	(1.0)	+	(0.95)	+
MAP3K2 (MEKK2)	+	(1.27)	−	(0.82)	−
MAPK1 (ERK2)	+	(1.03)	−	(0.76)	−
MAPK10	−	(0.55)	+	(1.24)	+
MAPK3 (ERK1)	+	(0.98)	−	(0.79)	+
MAPK8	+	(0.99)	−	(0.73)	+
MAPK9	unc.	(0.80)	−	(0.54)	+
MKNK1	−	(0.82)	−	(0.54)	+
RPS6KA5 (p70S6K)	+	(0.97)	unc.	(0.75)	+
RPS6KB1 (p70S6K)	+	(1.00)	−	(0.68)	+
TP53	+	(0.96)	unc.	(0.82)	+
DUSP1	+	(0.97)	−	(0.50)	+
DUSP6	+	(1.05)	−	(0.74)	+
IKBKB	+	(1.05)	−	(0.73)	+
PI3K signaling					
SHC1	+	(0.94)	−	(0.78)	+
GRB2	+	(1.07)	−	(0.76)	+
GAB1	+	(1.00)	+	(0.88)	+
P13KCA	+	(1.47)	−	(1.08)	−
PIK3R1	+	(0.98)	−	(0.80)	+
PIK3R2	+	(1.17)	−	(0.75)	−
E1F4E	+	(1.28)	+	(1.24)	−
PDPK1 (PDK1)	unc.	(0.95)	−	(0.86)	+
AKT1	+	(0.95)	−	(0.69)	+
AKT3	+	(1.11)	−	(0.87)	unc.
TP53	+	(0.96)	unc.	(0.82)	+
IKBKB (IKK)	+	(1.06)	−	(0.73)	+
GSK3A	unc.	(0.87)	−	(0.75)	−
GSK3B	+	(1.06)	−	(0.84)	unc.
RPS6KA5 (p70S6K)	+	(0.97)	unc.	(0.75)	−
RPS6KB1 (p70S6K)	+	(1.00)	−	(0.68)	−
NFKB	+	(1.02)	−	(0.90)	−
CCND1	−	(1.17)	−	(0.83)	−
JAK-STAT signaling					
SRC	+	(1.18)	+	(1.16)	+
STAT3	+	(1.06)	−	(0.89)	unc.
STAT5	+	(1.06)	−	(0.89)	−
JAK1	+	(1.02)	−	(0.77)	−
FOS	+	(1.50)	−	(1.02)	−
JUN	+	(1.32)	unc.	(0.98)	unc.
*ELK1*	+	(1.08)	−	(0.77)	−
PLC gamma signaling					
PLCG1 (PLC)	+	(1.18)	−	(0.89)	−
PPP2CA (TSC2)	+	(0.94)	−	(0.69)	−
PRKCA	+	(1.10)	−	(0.78)	unc.
RPS6KA5 (p70S6K)	+	(0.97)	unc.	(0.75)	−
RPS6KB1 (p70S6K)	+	(1.00)	−	(0.68)	−
IKBKB (IKK)	+	(1.06)	−	(0.73)	−
NFKB	+	(1.02)	−	(0.90)	−
Rho signaling					
RHOA	+	(1.08)	−	(0.84)	unc.

Values in the parenthesis show the fold change of a treatment medium against serum-free medium. + Maximum expression; − Minimum expression; unc. Unchanged.

## Data Availability

Not applicable.

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
