# Peer review of "Potential of Epidermal Growth Factor-like Peptide from the Sea Cucumber Stichopus horrens to Increase the Growth of Human Cells: In Silico Molecular Docking Approach"

_marinedrugs, 2022, doi:10.3390/md20100596_

Round 1
Reviewer 1 Report
The authors have identified the contig that that was retrieved from their transcriptomic data of sea cucumber (Stichopus horrenes) and predicted the protein structure based on the DNA contig and conducted the molecular docking with human epidermal growth factor receptor. The result demonstrated that Sh-EGF1-1 peptide activates EGFR pathway and induces cell proliferation via PI3K-AKT-GSK3, Ras- Raf-MEK-ERK-MAPK, PLC gamma, STAT and Rho signaling pathways. However, authors need to revise the bellows before publication in this journal.
- Characterization data for Sh-EGFl-1 should be added like synthesis process, amino acid sequence
- It should be clearly labeled whether Sh-EGFl-1 used in all experiments is a protein or a peptide. Difficulty in analyzing data and reading manuscripts.
- In Figure 3, the peptide numbering is different from the existing numbering, so it is difficult to interpret the data.
- Overall, to compare the structures of human EGF and Sh-EGFl-1 peptide, not only the 3D structure but also tabulated data should be submitted.
- There is no mention of a method for analyzing binding affinity.
- Although there is a separate discussion section, result is continuously included in the discussion about the result.
- Overall, the lack of interpretation of the data makes it difficult for readers to understand and the data image quality is also low. Also, the method needs to be accurately described and it would be good to clarify what the author intends to suggest.
- In the cell-related experiment qPCR, it is necessary to mention what the control is, and in the case of SFM, how was it marked on the heat map?
- Line 108: Write name of the cell line
- Line 281-283; authors are recommended to write statistical analysis section separately
- Fig. 6. How the authors count the cells? Did authors used analytical technique or microscopic observation. In addition, the discussion for the cell proliferation was not sufficient. Moreover, it seems there is no cell proliferation based on the figure. Also, it is recommended to use histogram instead of present figure.
- Line 300-304; How did the authors check the cell morphology. If used microscopy, why did not included images to the manuscript. Moreover, it seems the table 2 make reader confuse without cell images.
- Line- 475; Write more details on how the peptide was prepared.
Author Response
Thank you for your comments and suggestions.
Based on the your feedbacks on 16 August 2022, we have amended the manuscript accordingly.
Hopefully, this revised manuscript is improved and we are still open any other comments/suggestions. Your
thoughtful correspondences were very much appreciated and we hope manuscript will be considered for the
Marine Drug’s special issue Pharmaceutical, Nutraceutical, Cosmeceutical and Biotechnological Potentials of Southeast Asian Marine Resources.
Nurul Y MY

Reviewer 2 Report
The manuscript “Epidermal Growth Factor-like Peptide of Sea Cucumber Sti
chopus horrens Is Able to Increase Growth of Human CellsThrough EGF Pathway “is interesting and well written. The manuscript describes the molecular basis for the wound healing activity of the Sea Cucumber Stichopus horrens using extensive modeling and in silico study. It also describes the potential of the newly proposed peptide (i.e. Sh-EGFl-1) as a growth factor.
However, I have a number of comments that need to be addressed by the authors:
- In the results section, please add a new figure representing the whole sequence of Contig 498513 and the selected sequence Sh-EGFl-1. Also, add a 3d structure of the Contig 498513 protein labeling the selected sequence (i.e. Sh-EGFl-1) and the Ca2+ binding domains.
- Is the new modeled protein (i.e. Sh-EGFl-1) don’t need Ca2+ for stabilization? How to confirm that the newly modeled protein (i.e. Sh-EGFl-1) was stable enough to bind with the human EGFR. A molecular dynamics simulation run might be able to answer this question through investigation of the RMSD of the protein over at least 100 ns of dynamic simulation.
- It is not obvious how the authors construct the disulfide bridges of Sh-EGFl-1.
- Figure 2 needs to be zoomed in to show structure alignment details (Using pymol for this task is a good option)..
In the mutation study (Lines:206-213), the effect of mutation should be validated by running a molecular dynamics simulation.
The authors used pymol for visualization in figure 4, while they used Chimera in Figure 2. Please, use pymol for all visualizations to show more detailed structures.
The stability of the expressed peptide (i.e. Sh-EGFl-1) should be studied. I suggest that the authors perform the binding affinity study using the freshly prepared peptide, and another time with the stored peptide. To show that the peptide is stable and can preserve its binding affinity and to what extent?
In the methods section:
The modeling part is poorly described. Please, describe how you generate Sh-EGFl-1 model, and the docking step in more detail.
Author Response

(The authors gave the same response as above.)

Round 2
Reviewer 1 Report
The authors described "gene expression profiling of the cells indicated that Sh-EGFl-1 has activated hEGFR through five of epidermal growth factor signaling pathways; phosphoinositide 3-kinase (PI3K), mitogen-activated protein kinase (MAPK), phospholipase C gamma (PLC-gamma), Janus kinase-signal transducer and activator of transcription (JAK-STAT) and Ras homologous (Rho) pathways".
However, it should be verified with clear data for RNA-level and protein-level expression. In detail, the authors should show essential results for specific marker expression and protein production on the EGF signaling pathways in the cells with Western blot assay and ELISA kits.
Author Response
Dear Reviewer 1,
Thank you for your effort and prompt action to make possible for a timely efficient review process for our manuscript.
Before you make the final dicision, let me share our motivation statement for this manuscript:
Although many research have been conducted, there are still lacking of scientific proof to relate the therapeutic effect of sea cucumber’s product in wound healing. Most of the previous studies particularly performed by the local scientist are superficial by only looking at the effect of the sea cucumber extract on cell culture.
Thus, it is our main objective to investigate deeper into looking what are those specific biomolecules in the sea cucumber that could contribute to its therapeutic claims and its capability to self-regenerate. At a big picture, our research strategy is to use genomics data to pinpoint potential molecules that could be further elucidated using in silico approach and subsequently to validate the outcome by wet lab analyses. The cell-ligand binding assay is to proof the real binding between the Sh-EGFl-1 and human cell’s EGFR. As the interaction occurs, it is interesting to know whether this would lead to either mimicking or blocking effect on the expected pathway that could be verified at the molecular level that we chose the qPCR or real time-PCR array approach to get a gene (RNA) profile of the involved proteins.
Based on my previous works, qPCR or real time-PCR is the most reliable method to verify large scale RNA analysis such as RNA microarray and transcriptomic. We also used qPCR or real time-PCR to validate our RNA based-genosensor for viral detection.
Therefore, the following is our feedback to the comment and suggestion from Reviewer 1 in the second round review:
Comment & Suggestion from Reviewer 1 (2nd round review - 8 September 2022)
The authors described "gene expression profiling of the cells indicated that Sh-EGFl-1 has activated hEGFR through five of epidermal growth factor signaling pathways; phosphoinositide 3-kinase (PI3K), mitogen-activated protein kinase (MAPK), phospholipase C gamma (PLC-gamma), Janus kinase-signal transducer and activator of transcription (JAK-STAT) and Ras homologous (Rho) pathways".
However, it should be verified with clear data for RNA-level and protein-level expression. In detail, the authors should show essential results for specific marker expression and protein production on the EGF signaling pathways in the cells with Western blot assay and ELISA kits.
It is understood that proteins should be the final product of expressed genes and could be differently represented due to post-transcription or post-translation processes, which we really appreciate your suggestions to include protein analysis such as Western blot or ELISA in our study.
However, due to high amount of time spent on the works described in the manuscript, we decided to publish the genomics à in silico prediction à validation through cell assays (ligand binding & cell proliferation) à pathway analysis using qPCR array as our main research work that hopefully could contribute and add value of the scientific information of this sea cucumber species.
If it is concern that the title “Epidermal Growth Factor-like Peptide of Sea Cucumber Stichopus horrens Is Able to Increase Growth of Human Cells Through EGF Pathway” could be misleading and too specific due to verification of the effect is only at the RNA level, we would suggest to change the manuscript title to “Potential of Epidermal Growth Factor-like Peptide from Sea Cucumber Stichopus horrens to Increase Growth of Human Cells: In Silico Molecular Docking Approach”. Hopefully this title will represent this research paper that focus on predicting the Sh-EGFl-1:hEGFR interaction that subsequently validated by cell based-assay and qPCR based-pathway analysis.
We are planning to conduct more experiments in our future work particularly using other binding assays (such as labelled-interaction microscopic analysis and Biocore’s binding tool) and verification at protein level using large scale protein analysis such as proteomic to more conclusive data on this particularly, Sh-EGFl-1:hEGFR interaction, and other similar findings.
We anticipate that our sharing through this manuscript would give scientific meaning for other scientist to explore the potential of this unique marine species in many applications.
Thank you
Reviewer 2 Report
The authors addressed most of my concerns. I think the paper is now eligible to be published.
Author Response
Thank you for your effort and prompt action to make possible for a timely efficient review process for our manuscript.
Your support in our research work is very much appreciated.
Round 3
Reviewer 1 Report
The authors did properly change the title to focus on in silico study. Thus it is acceptable for publication in this journal
Author Response
Dear Reviewer,
Thank you very much for your support.